# Characterization and Genomic Analyses of dsDNA Vibriophage vB_VpaM_XM1, Representing a New Viral Family

**DOI:** 10.3390/md22090429

**Published:** 2024-09-21

**Authors:** Zuyun Wei, Xuejing Li, Chunxiang Ai, Hongyue Dang

**Affiliations:** 1College of Ocean and Earth Sciences, Xiamen University, Xiamen 361102, China; weizuyun@stu.xmu.edu.cn (Z.W.); lixuejing1989@xmu.edu.cn (X.L.); 2State Key Laboratory of Marine Environmental Science, Xiamen 361102, China; 3Fujian Key Laboratory of Marine Carbon Sequestration, Xiamen 361102, China; 4State Key Laboratory of Mariculture Breeding, Xiamen 361102, China

**Keywords:** *Vibrio parahaemolyticus*, bacteriophage, genome analysis, new viral family

## Abstract

A novel vibriophage vB_VpaM_XM1 (XM1) was described in the present study. Morphological analysis revealed that phage XM1 had *Myovirus* morphology, with an oblate icosahedral head and a long contractile tail. The genome size of XM1 is 46,056 bp, with a G + C content of 42.51%, encoding 69 open reading frames (ORFs). Moreover, XM1 showed a narrow host range, only lysing *Vibrio xuii* LMG 21346 (T) JL2919, *Vibrio parahaemolyticus* 1.1997, and *V. parahaemolyticus* MCCC 1H00029 among the tested bacteria. One-step growth curves showed that XM1 has a 20-min latent period and a burst size of 398 plaque-forming units (PFU)/cell. In addition, XM1 exhibited broad pH, thermal, and salinity stability, as well as strong lytic activity, even at a multiplicity of infection (MOI) of 0.001. Multiple genome comparisons and phylogenetic analyses showed that phage XM1 is grouped in a clade with three other phages, including *Vibrio* phages Rostov 7, X29, and phi 2, and is distinct from all known viral families that have ratified by the standard genomic analysis of the International Committee on Taxonomy of Viruses (ICTV). Therefore, the above four phages might represent a new viral family, tentatively named *Weiviridae*. The broad physiological adaptability of phage XM1 and its high lytic activity and host specificity indicated that this novel phage is a good candidate for being used as a therapeutic bioagent against infections caused by certain *V. parahaemolyticus* strains.

## 1. Introduction

*Vibrio parahaemolyticus* is a Gram-negative halophilic bacterium widely distributed in estuarine, coastal, and marine environments [1,2,3]. This bacterium is predominantly associated with various edible sea animals, including fish and shellfish such as shrimp, lobsters, crabs, and oysters [4]. *V. parahaemolyticus* has also been found to be the causative agent of acute gastroenteritis in humans resulting from the consumption of undercooked or raw seafood [5,6]. In China, *V. parahaemolyticus* has been a leading cause of foodborne disease outbreaks and infectious diarrhea cases in coastal areas [7]. 

Bacteriophages (phages) are viruses that specifically infect bacteria and represent the most abundant biological entity on the planet. They can kill nearly half of the bacterial population every two days and play a critical role in bacterial control in the natural environment [8,9,10]. Bacteriophages are highly diverse morphologically and genetically, and they can facilitate horizontal gene transfer. Therefore, they are crucial for bacterial diversity and evolution [11]. Furthermore, phage genomes contain many new genes with unknown functions, so they may be one of the largest unexplored gene pools [12,13].

Antibiotic resistance has increased significantly due to widespread antibiotic abuse, leading to the emergence of multidrug-resistant bacteria in marine aquaculture and natural environments, causing the spread and therapeutic difficulty of bacterial pathogens [14,15,16]. Phage therapy has been considered a promising method to control antibiotic-resistant bacterial pathogens [17,18,19,20]. Bacteriophages with high host specificity, high lytic activity, and eco-friendly properties are beneficial candidates as biocontrol agents [21,22,23]. Indeed, specific phages have been successfully applied as biocontrol agents to control foodborne pathogens [24,25]. However, resources on phages are still very minimal. As of 30 June 2023, only 122 Vibrio and 26 *V. parahaemolyticus* phage genomes were recorded in the NCBI RefSeq database (http://www.ncbi.nlm.nih.gov/genome, accessed on 30 June 2023). It is necessary to isolate and characterize new phages to broaden our understanding of the ecology, evolution, and diversity of both phages and their bacterial hosts further. 

Bacteriophage classification and genetic backgrounds are critical for the application of phage therapy [26,27]. In earlier studies, phages were classified mainly according to their morphological similarity and nucleic acid composition [28]. Specific conserved genes of phages were also used for phage taxonomic analyses, such as those encoding the large subunit of terminases and the major capsid proteins [29]. Sequencing technology is becoming more advanced, and phage taxonomic classifications based on genomes, transcription mechanisms, and gene contents are becoming more accurate [30]. Based on the virus taxonomic classification by the International Committee on Taxonomy of Viruses (ICTV), *Duplodnaviria* currently contains one kingdom (*Heunggongvirae*), two phyla (*Peploviricota* and *Uroviricota*), two classes (*Herviviricetes* and *Caudoviricetes*), eight orders, and sixty-six families, including those that do not belong to any defined orders. Tailed phages all belong to class *Caudoviricetes* according to the latest nomenclature rules of the ICTV. In contrast, they had previously been classified as *Myoviridae*, *Podoviridae*, and *Siphoviridae* based on their tail morphology. With the development of genome sequencing and phylogenetic analysis, phage taxonomy has changed [31]. At the time of writing, the ICTV has abolished the phage nomenclature of *Myoviridae*, *Podoviridae*, and *Siphoviridae*.

The present study reported a new vibriophage isolated from *V. parahaemolyticus* (i.e., vB_VpaM_XM1). The morphology, host range, one-step growth curve, stabilities against pH, salinity, and thermal changes of vB_VpaM_XM1 were evaluated. Based on genomic annotation and comparative genomic and phylogenetic characterizations, vB_VpaM_XM1 and three other phages represent a new viral family.

## 2. Results and Discussion

### 2.1. Biological Characterization of XM1

The phage vB_VpaM_XM1 (XM1 in short) was isolated from the temporary maintenance water of marketed marine fish using *V. parahaemolyticus* as the host bacteria. It was able to form clear, circular, and boundary-smooth plaques (Figure 1A). As shown in Figure 1B, transmission electron microscope (TEM) analysis revealed that XM1 carries an icosahedral head (76.92 ± 2.65 nm long and 64.10 ± 1.36 nm wide) and a long contractile tail (130.8 ± 5.7 nm).

The lytic cycle of XM1 was determined with a one-step growth curve at a 0.01 multiplicity of infection (MOI). The latent period of phage XM1 was about 20 min, and its burst size was approximately 398 PFU/cell (Figure 2A). Previously, phage vB_VPAP_DE10 infecting *V. parahaemolyticus* had been shown to have a latent period of approximately 0~25 min, with a burst size of 19 PFU/cell [32]. Phages F23s2 and H256D1 showed a latent period of 0–20 min and 0–5 min, respectively, with a burst size of 12 PFU/cell and 131 PFU/cell, respectively [33]. *Vibrio* phage VP06 has a latent period of 30 min and a burst size of 60 PFU/cell [34]. Compared with these previously studied *V. parahaemolyticus*-infecting phages, phage XM1 exhibited a similar latent period and a much larger burst size.

To examine the host range of phage XM1, a spotting test was performed against 58 bacterial strains isolated from various environments (Appendix A). The results showed that XM1 only lyses *V. xuii* LMG 21346 (T) JL2919, *V. parahaemolyticus* 1.1997, and *V. parahaemolyticus* MCCC 1H00029, suggesting that XM1 has a narrow range of and high specificity to its hosts. 

Furthermore, the effects of pH, temperature, and salinity on the stability of phage XM1 were tested. XM1 maintained activity from pH 4 to 10, with the highest lytic activity at pH 9 (Figure 2B). The optimal pH for *V. parahaemolyticus* phages CA8 and BA3 was from pH 5 to 7 and pH 6 to 7, respectively [12], while *V. parahaemolyticus* phage R18L exhibited stability from pH 6 to 11 [35]. Thermal stability tests showed that phage XM1 was stable at 4 °C to 60 °C for 3 h with a decreasing stability trend with increasing temperature and a total loss of activity at 70 °C (Figure 2C). Phages CA8 and BA3 were stable only at temperatures ranging from 20 °C to 40 °C [12], while phage R18L was stable from 4 °C to 40 °C [35]. Phage XM1 in SM buffer at 3% salinity showed the most significant activity but could not grow at 0% salinity (Figure 2D), indicating coastal and marine environments as XM1’s habitat. *V. parahaemolyticus* phage VB_VpP_BT-1011 can survive at 0% to 3% NaCl [36], indicating this phage also includes estuarine environments as its habitat. The results of the current study show that phage XM1 is stable at broad pH, temperature, and salinity ranges, indicating its tolerance and adaptation to various environmental stresses.

*V. parahaemolyticus* was infected with XM1 at different MOIs to investigate its effect on bacterial growth. The inhibition curve (Figure 3) indicated that increasing the MOI increased bacterial growth inhibition. There was an initial increase in the concentration of host bacteria. However, with the release of phages, the concentration of host bacteria began to decrease, with no observable difference in XM1’s inhibitory effects observed after 11 h of infection, regardless of different MOIs. This intense bactericidal activity indicates that XM1 is a potent candidate for use in phage therapy.

### 2.2. Genome Sequence of Vibrio Phage XM1

Genome analyses revealed that Phage vB_VpaM_XM1 (Accession: PP580404) is a double-stranded DNA virus belonging to the *Duplodnaviria* realm in the ICTV. Its genome size is 46,056 bp with a total G + C content of 42.51%, containing 69 predicted open reading frames (ORFs). The size of XM1 protein-coding sequences (CDSs) ranges from 51 to 825 amino acid residues (208 on average). Among the 69 ORFs, 66 are transcribed in the forward direction, while the other 3 (i.e., *ORF4*, *ORF35*, and *ORF36*) are transcribed in the reverse direction. The 69 predicted genes primarily encode viral structure proteins and proteins for DNA packaging, DNA metabolism and replication, and host lysis (Figure 4). Nearly two-thirds of the predicted genes (45/69) can be assigned functions according to their homology to known sequences of other phages. Concretely speaking, 19 predicted genes (*ORF5*, *ORF8*, *ORF9*, *ORF11*, *ORF12*, *ORF13*, *ORF15*, *ORF16*, *ORF18*, *ORF19*, *ORF20*, *ORF21*, *ORF22*, *ORF23*, *ORF24*, *ORF26*, *ORF27*, *ORF29*, and *ORF30*) are related to viral structure proteins, 3 predicted genes (*ORF2*, *ORF3*, and *ORF6*) are associated with DNA packaging, and 14 predicted genes (*ORF1*, *ORF7*, *ORF10*, *ORF37*, *ORF40*, *ORF41*, *ORF42*, *ORF44*, *ORF46*, *ORF48*, *ORF56*, *ORF58*, *ORF59*, and *ORF64*) are connected with DNA replication and metabolism. 

On the other hand, the remaining predicted genes (24 ORFs) encode hypothetical proteins with unknown functions. Specifically, protein sequences encoded by *ORF4*, *ORF51*, *ORF60*, and *ORF66* show no homology to any known protein sequences in the database, potentially indicating their sequence and functional novelties. The predicted terminase large subunit protein (*ORF3*) plays an important role in the late stage of viral DNA packaging [37]. The predicted portal protein (*ORF6*) is involved in dsDNA viral genome packaging and release [38]. Three ORFs are predicted to encode host lysis proteins, including a lysozyme (encoded by *ORF25*), a sporulation-specific N-acetylmuramoyl-L-alanine amidase (encoded by *ORF32*) that belongs to the families of peptidoglycan hydrolases [39], and a Rz-like spanin (encoded by *ORF36*) that interacts with bacterial outer membrane and plays a role in the final step of host lysis [40]. Additionally, *ORF45* encodes a site-specific integrase that may be a lysogen-related protein [41]. Moreover, no virulence gene or factor was found in the genome of phage XM1. Therefore, phage XM1 may be safe for use in phage therapy.

### 2.3. Phylogenetic and Comparative Genomic Analyses of Phage XM1

To determine the phylogenetic taxonomy of phage XM1, a proteomic tree based on viral whole genomes was generated using VipTree (https://www.genome.jp/viptree, accessed on 10 June 2023) [42]. The result shows that XM1 is closely clustered with *Vibrio* phages Rostov 7 (accession: MK575466.1) [43], X29 (accession: NC_024369.2) [44], and phi2 (accession: KJ545483.2) [45] (Figure 5A,B), indicating that these four phages may form a new taxonomic group of viruses. 

According to BLASTN in the NCBI database, phage XM1 has the highest genomic sequence identity to *Vibrio* phages Rostov 7, X29, and phi2, with 77.76%, 77.60%, and 77.54% identity scores and 33%, 25%, and 25% query coverage, respectively (accessed on 10 June 2023). The ANI value of the XM1 genome with the viral genome of Rostov 7, X29, and phi 2 was 73.35%, 72.22%, and 72.23%, respectively (Figure 6A). However, phage XM1 showed no genomic match with other NCBI viral genomes. To further confirm the similarity of phage XM1 to other phage genomes, intergenomic similarity value analysis was performed using VIRIDIC v1.1. As shown in Figure 6B, the intergenomic similarity values of the XM1 genome with the viral genome of Rostov 7, X29, and phi 2 were 55.1%, 56.4%, and 56.5%, respectively. Phage XM1 shares a similar overall genomic organization with *Vibrio* phages Rostov 7, X29, and phi 2 (Figure 6C). The predicted genes of XM1 showed 66% to 100% sequence identities to genes of the other three *Vibrio* phages, except several genes that had no sequence match.

To further confirm the taxonomic novelties of phage XM1 and its closest relatives (i.e., *Vibrio* phages Rostov 7, X29, and phi 2), a whole-genome phylogenetic tree (Appendix A, Figure 7) was constructed with 154 representative viral genomes selected from all the 66 families of *Duplodnaviria* currently defined by the ICTV (accessed on 9 June 2023). The phylogenetic tree showed that phages Rostov 7, X29, and phi 2 belong to *Duplodnaviria*, *Heunggongvirae*, *Uroviricota*, and *Caudoviricetes*. In addition, the analytic result indicated that phages XM1, Rostov 7, X29, and phi 2 are phylogenetically grouped and form a unique viral cluster not affiliated with any known viral families. Therefore, we tentatively propose *Weiviridae* as the new family name for these four novel *Vibrio*-lysing phages.

The terminase large subunit is a relatively conserved protein used as a marker for establishing phage phylogenetic relationships [46]. Similarly, the major capsid protein, the primary component of the phage capsid, is conserved among phylogenetically related phages and is frequently used in phage classification [47]. In this study, a phylogenetic tree was constructed based on the protein sequences of phage terminase large subunit and phage capsid protein, respectively (Figure 8). Our analyses show that phage XM1 is grouped with *Vibrio* phages Rostov 7, X29, and phi 2 and phylogenetically distant from other phage families. The genome phylogenetic tree also showed similar results (Appendix A).

The gene sequences selected in these two phylogenetic trees were based on Blast in NCBI (Accessed on 10 June 2023). Values at the nodes indicate the bootstrap support calculated from 1000 replicates. 

According to phylogenetic trees (Figure 5, Figure 7 and Figure 8), the *Vibrio* phages Rostov 7, X29, and phi 2 belong to class *Caudoviricetes*. However, none of them are classified into any known phage families. Phage XM1 and these phages are grouped as a new clade and different from previously described phages. These results indicate that phages XM1, Rostov 7, X29, and phi 2 can be classified as a new phage family.

## 3. Materials and Methods

### 3.1. Phage Isolation and Purification

*V. parahaemolyticus* 1.1997 was used as the bacterial host [48]. It was grown in a rich organic (RO) medium with a shaking speed of 160 rpm/min at 28 °C. Firstly, 1 mL of water sample from the eighth seafood markets (Xiamen, China) was added into 10 mL of an exponentially growing culture of *V. parahaemolyticus* 1.1997 and incubated for 24 h. The mixed culture was then passed through a 0.22 μm filter membrane (Millipore, Bedford, MA, USA) to remove bacterial cells. The filtrate was diluted and mixed with exponential host cultures to obtain the phage plaque using the double-layer agar method [49]. After the above-mentioned steps, the well-separated plaque was removed and stored in the storage medium (SM) buffer (50 mM Tris-HCl, 0.1 M NaCl, and 8 mM MgSO_4_, pH 7.5) at 4 °C for later use.

### 3.2. Phage Enrichment

To obtain a highly concentrated phage, 1 L phage lysate was treated with DNase I and RNase A at room temperature for 1 h until the final concentration reached 1 μg/mL. Then, 1 M NaCl was supplied for 30 min at 4 °C to promote the separation of phage particles and cell debris. Finally, the solution was mixed with 10% polyethylene glycol (PEG 8000, LABLEAD, Beijing, China) and stored for 3 d at 4 °C to precipitate virions. Viral particles were subsequently collected by centrifugation (12,000× *g*, 60 min, 4 °C) and resuspended in 6 mL of SM buffer. The phage suspensions were prepared via cesium chloride gradient centrifugation (1.3, 1.5, 1.7 g/mL) and centrifuged at 200,000× *g* for 24 h at 4 °C using an Optima L-100 XP ultracentrifuge (Beckman Coulter, CA, USA). The visible phage band was extracted and later dialyzed through 30 kD super-filters (Millipore, Bedford, MA, USA) [50].

### 3.3. Morphology Observation

Phage morphology was observed using a JEM-2100 transmission electron microscope (JEOL, Tokyo, Japan) at an acceleration voltage of 80 kV. To prepare the samples for observation, 20 μL of high-titer phage concentrate was plated on 200-mesh formvar-coated copper electron microscope grids and allowed to absorb for 10 min, then negatively stained with 1% phosphotungstic acid for 1 min, followed by air drying for 10 min. The size of phage particles was measured from at least five TEM images using ImageJ software (V1.8.0) [51].

### 3.4. Host Range

The host range of XM1 was determined by spot testing and confirmed by the double-layer agar method [52]. First, 1 mL of exponentially growing bacteria (10^8^ CFU/mL) was mixed with 5 mL of the pre-warmed (50 °C) semisolid liquid medium, then poured onto a solid agar plate immediately. After 10 min of air drying, 5 μL of purified phage solution was spotted on the host bacterial lawn. The plate was then incubated at 28 °C for 24 h. Phage infection was determined by visual examination of the plates for plaques. The used bacteria included 58 strains in the genera *Vibrio*, *Idiomarina*, *Pseudoalteromonas*, *Photobacterium*, and *Shewanella* (listed in Appendix A).

### 3.5. One-Step Growth Curve

The one-step growth curve of phage XM1 was determined using the previously described method [53]. Briefly, 1 mL of exponentially growing bacteria (10^8^ CFU/mL) was exposed to phages at a MOI of approximately 0.01, then placed in the dark for 10 min. Bacteria were then pelleted (6,000× *g*, 5 min), and the non-adsorbed phages in the supernatant were discarded. The pellet was then washed twice and resuspended in 100 mL RO medium, and the culture was then incubated at 28 °C with a shaking speed of 160 rpm/min. Every 10 min, subsamples were collected, and the viral abundance was detected using the double-layer agar method. The burst size was calculated as the ratio between the number of virions at the growth plateau and the initial number of infected host cells [54].

### 3.6. pH, Temperature, and Salinity Tolerance

A series of 3 experiments were designed to determine the influence of pH, temperature, and salinity on the stability of phage XM1. In all experiments, the double-layer agar method was applied to estimate the infection activity of the phage. In the pH experiment, the pH of the SM buffer was adjusted from 2 to 12 with HCl or NaOH solution. The phage concentrate was added to the SM buffer so that the final concentration was 10^14^ PFU/mL, and then all treatments were incubated at 4 °C for 3 h, 24 h, and 48 h. For the experiment that investigated the thermal stability of the phage, the phage in all treatments was incubated for 3 h, with incubation temperatures set at 4 °C, 24 °C, 37 °C, 50 °C, 60 °C, and 70 °C. As for the salinity tolerance experiment, solutions with salinity ranging from 0 to 5% were used for phage incubation (incubation time: 12 h).

### 3.7. Growth Curve Experiment 

The phage XM1 was mixed with the host *V. parahaemolyticus* at different MOIs (0.001, 0.01, 0.1, 1, 10) and incubated at 28 °C. Meanwhile, *V. parahaemolyticus* at the same MOI level but without the phage was used as a positive control. The growth curves were monitored over 12 h, and optical density (OD_600_) measurements were recorded every 1 h. Three independent assays were carried out for each assay.

### 3.8. DNA Extraction, Genome Sequencing, and Genome Assembly

Viral genomic DNA was extracted using the TakaRa MiniBEST Viral RNA/DNA Extraction Kit according to the manufacturer’s protocol. In brief, 200 μL viral concentrate was mixed with 200 μL Buffer VGB, 20 μL Proteinase K, and 1 μL Carrier RNA, then incubated at 56 °C for 10 min. After that, 200 μL of ethanol was added to the mixture before a 2-min centrifugation (12,000× *g*). Next, 500 μL RWA was added, and the solution was centrifuged at 12,000× *g* for 1 min. Following that, 700 μL RWB was added and the mixture was centrifuged at 12,000× *g* for 1 min, and this step was repeated twice. Finally, 30 μL RNase-free dH_2_O was added into the centrifuge tube and incubated for 5 min at room temperature before the final centrifugation was conducted (12,000× *g* for 2 min). The extracted DNA was stored at −20 °C. Phage genome sequencing was performed using the Illumina Nova platform by Shanghai Hanyu Bio-Tech Co., Ltd. (Shanghai, China). Then, the phage genome was assembled using velvet v1.2.03/Newbler v2.8/SOAPdenovo2 v2.04.

### 3.9. Genome Annotation and Phylogenetic Analysis

The open reading frames (ORFs) of the XM1 genome were predicted by the Glimmer3 v3.02/GeneMarkS v4.28/Prodigal v2.60 online server and annotated by a BLASTp search against the National Center for Biotechnology Information (NCBI) nonredundant (nr) protein sequences (accessed on 27 Feb 2023) [55,56]. A gene map was created based on the genome annotations using CGView-Circular Genome Viewer (https://proksee.ca, accessed on 27 Feb 2023) [57]. Genomic structures and comparison maps of phages belonging to the same categories were made using EasyFig v2.2.5 [58].

A phylogenetic tree based on genome sequence similarities computed by tBLASTx was constructed using the Viral Proteomic Tree server (VipTree, https://www.genome.jp/viptree/, accessed on 10 June 2023) [42]. OrthoFinder was used to compare the genomic similarity by orthology (OrthoANI v0.93.1), which was calculated using the BLASTp analysis [59]. The intergenomic similarities was calculated by Virus Intergenomic Distance Calculator (VIRIDIC v1.1) (https://rhea.icbm.uni-oldenburg.de/VIRIDIC/, accessed on 1 August 2024). To explore the phage’s taxonomic status, the complete nucleotide sequence of phage XM1 and its related viral genomic sequences were submitted to the virus classification and tree building online resource (VICTOR) (http://ggdc.dsmz.de/victor.php, accessed on 9 June 2023) for phylogenetic analysis, with the recommended settings of genome BLAST distance phylogeny (GBDP) method being used [60]. The terminase large subunit protein and capsid protein sequences of XM1 were used to construct phylogenetic trees to analyze its evolutionary relationships, and a Neighbor-joining method in the MEGA 6.0 software package with 1000 bootstrap replicates was used to construct the phylogenetic tree (accessed on 10 June 2023) [61,62].

## 4. Conclusions

This study isolated and fully characterized a new phage, vB_VpaM_XM1, which infects *V. parahaemolyticus* and has a large burst size and a narrow host range. XM1 has a broad range of temperature, pH, and salinity adaptability and exhibits strong lytic activity. These results indicated that XM1 has great potential as a novel antibacterial agent for the biological control of vibriosis in aquaculture. Moreover, the complete XM1 genome sequence was determined and compared with its phage relatives. Furthermore, phylogenetic analyses revealed that XM1 clusters a new clade with *vibrio* phages Rostov 7, X29, and phi 2 and should belong to a new viral family named *Weiviridae*. Our report provides an in-depth analysis of the phage at the genomic, phylogenetic, and ecological levels and provides a potential antimicrobial candidate for pathogenic *V. parahaemolyticus*.

## Figures and Tables

**Figure 1 marinedrugs-22-00429-f001:**
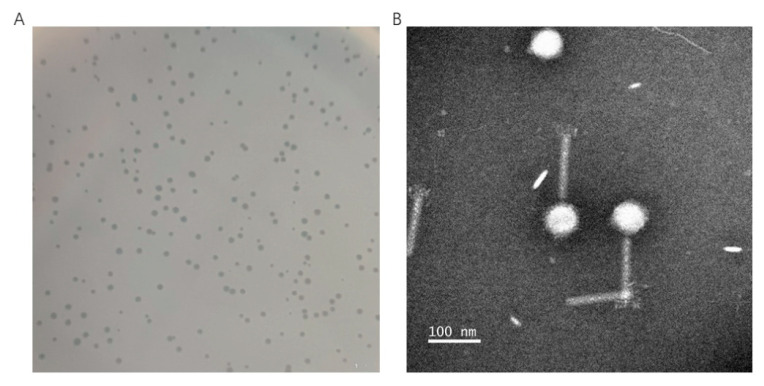
Morphology of phage vB_VpaM_XM1. (**A**) Plaques of vB_VpaM_XM1 infecting *V. parahaemolyticus* 1.1997. (**B**) Transmission electron micrograph of vB_VpaM_XM1. The scale bar represents 100 nm.

**Figure 2 marinedrugs-22-00429-f002:**
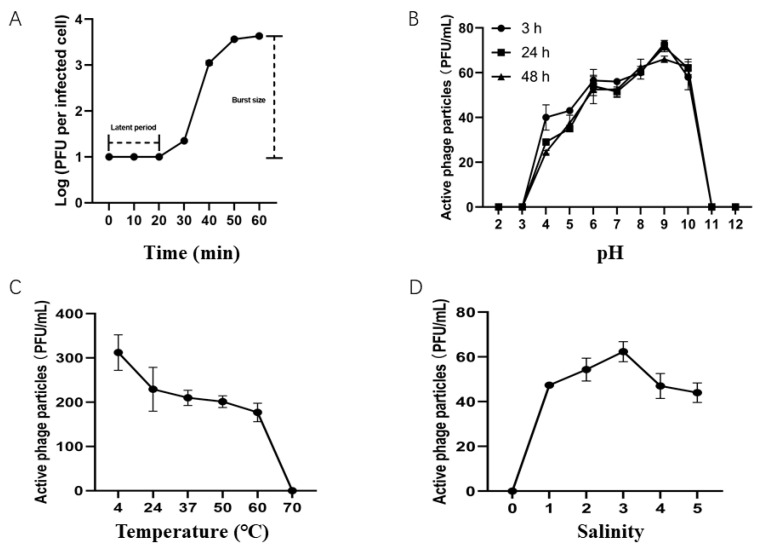
Biological properties of phage vB_VpaM_XM1. (**A**) One-step growth curve of phage vB_VpaM_XM1. (**B**) pH stability curve of phage vB_VpaM_XM1. (**C**) Stability of phage vB_VpaM_XM1 in different temperatures. (**D**) Stability of phage vB_VpaM_XM1 in different salinity. The data shown are average values from triplicate experiments, and error bars indicate standard deviations.

**Figure 3 marinedrugs-22-00429-f003:**
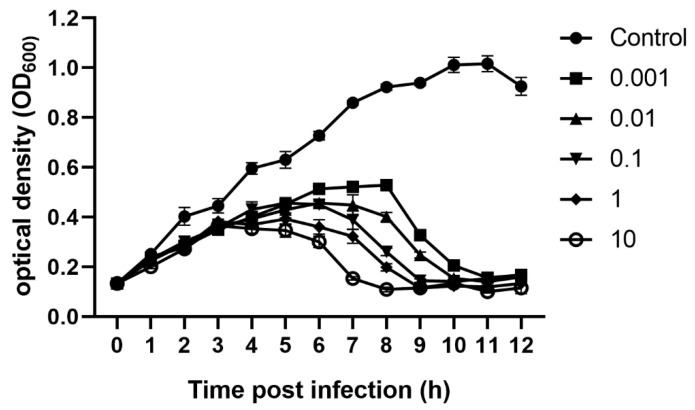
Inhibition curves of *V. parahaemolyticus* by phage vB_VpaM_XM1 at various MOIs (0.001, 0.01, 0.1, 1, and 10).

**Figure 4 marinedrugs-22-00429-f004:**
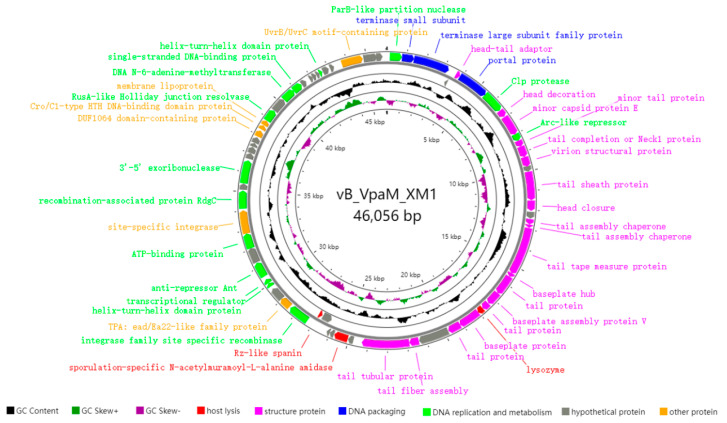
Annotated genome map of phage vB_VpaM_XM1. The 69 ORFs are represented by colored arrows, and the direction of each arrow represents the direction of transcription.

**Figure 5 marinedrugs-22-00429-f005:**
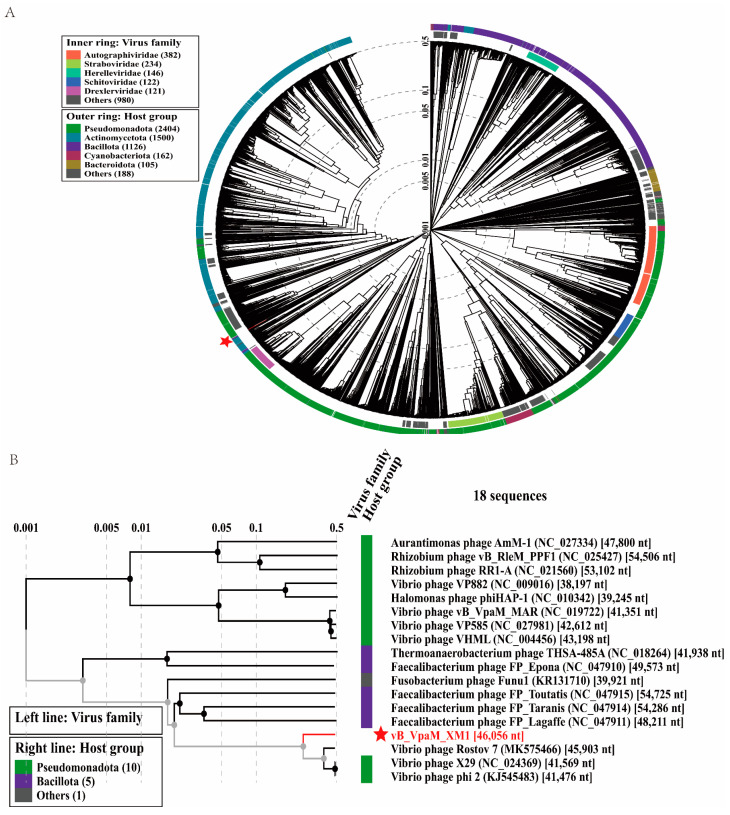
Phylogenetic analyses of phage vB_VpaM_XM1. (**A**) A circular proteomic tree constructed with phage vB_VpaM_XM1 and other phage genomic sequences using VipTree. (**B**) The viral proteomic tree, including vB_VpaM_XM1 and its 17 nearest phage relatives. The phages selected were a part of a rectangular tree of the whole genome. The left color line indicates the viral taxonomic families (the left color line is blank because there is no specific virus family for these viruses), and the right color line indicates the host groups.

**Figure 6 marinedrugs-22-00429-f006:**
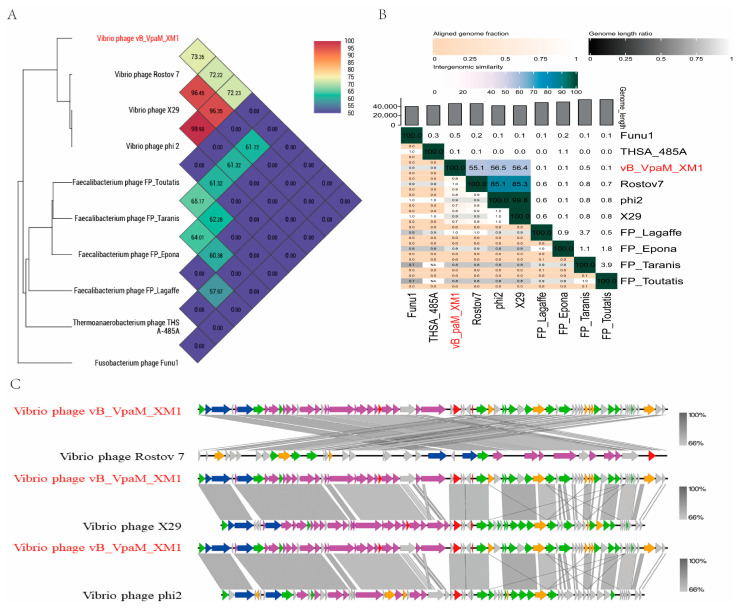
Comparative genomic analyses of phage vB_VpaM_XM1. (**A**) Genome-wide tree based on the average nucleotide identity (ANI) from 10 phages. Ten phages were selected based on the phages that showed the closest relationships to vB_VpaM_XM1 in the evolutionary tree in Figure 5. (**B**) VIRIDIC-generated heatmap incorporating intergenomic similarity values (right half) and alignment indicators (left half and top annotation). (**C**) Genome organization and comparisons of phage vB_VpaM_XM1 with *Vibrio* phage Rostov 7, *Vibrio* phage X29, and *Vibrio* phage phi2. ORFs are depicted by leftward- or rightward-oriented arrows according to the direction of transcription. Each color indicates a putative function, including host lysis (red), DNA packaging (blue), DNA replication and metabolism (green), structure protein (purple), other protein (orange), or hypothetical protein (gray).

**Figure 7 marinedrugs-22-00429-f007:**
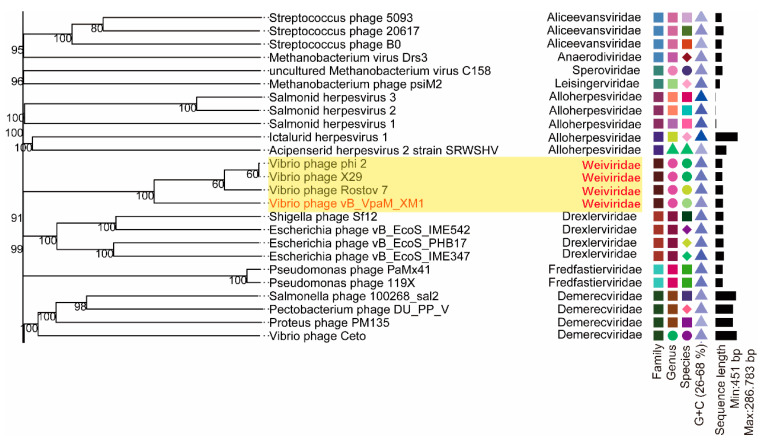
Local details of the Genome BLAST distance phylogeny (GBDP) tree constructed for *Vibrio* phages XM1, Rostov 7, X29, and phi 2 and 154 other viruses representing all the 66 known families in the realm *Duplodnaviria*. The truncated phylogenetic tree shows that vibrio phages XM1, Rostov 7, X29, and phi 2 are phylogenetically grouped and form a unique viral cluster unaffiliated with any known viral families in *Duplodnaviria*. The new phage family is tentatively named *Weiviridae*. The complete GBDP tree is shown in Appendix A. Numbers at the nodes are GBDP pseudo-bootstrap values (100 replications and values > 50%). The different colors and shapes represent different Family, Genus, Species and G + C content.

**Figure 8 marinedrugs-22-00429-f008:**
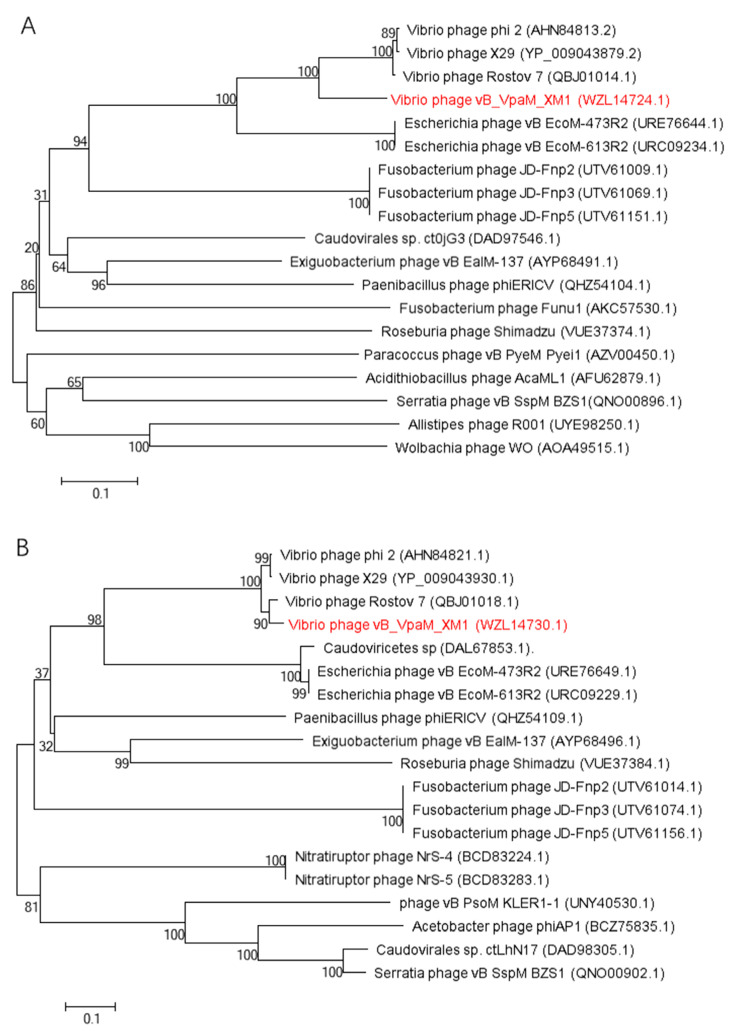
Neighbor-joining phylogenetic trees of phage vB_VpaM_XM1. (**A**) Phylogenetic tree based on the amino acid sequence of the terminase large subunit. (**B**) Phylogenetic tree based on the major capsid protein, showing the relationships between phage vB_VpaM_XM1 and other nearest phages.

## Data Availability

The authors declare that all relevant data supporting the findings of this study are available within the article and its Appendix A. The whole genomes of vB_VpaM_XM1 was deposited in NCBI database with accession number PP580404.

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
