# Peer review of "Characterization and Genomic Analyses of dsDNA Vibriophage vB_VpaM_XM1, Representing a New Viral Family"

_marinedrugs, 2024, doi:10.3390/md22090429_

Round 1
Reviewer 1 Report
Comments and Suggestions for Authors
This is a well-written manuscript that describes a newly isolated, unique bacteriophage that infects a few Vibrio isolates.
Specific comments
Line 82 Your measurements are not accurate to 4 significant figures. Please round off to the nearest nm.
Lines 89-90 If the latent period is 40 min, then calculate the burst size based on the 60 min time point. At 150 min there would be enough time for 2-3 rounds of phage growth.
Line 137 significant figures! “208 on average”
Fig. 5B Please enlarge to increase readability.
Reviewer 2 Report
Comments and Suggestions for Authors
The manuscript by Zuyun Wei describes a new vibriophage, vB_VpaM_XM1. The authors isolated the phage, contacted a number of biological experiments, sequenced the phage's genome and characterised it. Based on the ANI calculations and phylogenetic analysis, the authors proposed a new family, Weiviridae. Based on the results of genome analysis and basic biological experiments, the authors proposed phage vB_VpaM_XM1 as a good candidate for use in phage therapy.
The design of the study is clear, the figures are illustrative, and the introductory and discussion sections contain some interesting moments. However, the main conclusions regarding the proposed classification and the possibility of its use in phage therapy are not supported by the results:
1. The phage is temperate and it has a very narrow host range. It can not be “a good candidate for being used as a therapeutic bioagent against infections”.
2.ANI calculations show a level of similarity between the four Vibrio phages of over 70%. This means that these phages may form a genus but not a family. Furthermore, these calculations show a suspiciously high level of similarity (about 61%) between Vibrio phage Rostov 7 and the Faecaliumbacerium phage FP_Taranis.
Other issues:
Line 11 - Please replace “Myoviridae-like” with “myovirus”
Line 22 and elsewhere - please use italics in the names of viral taxa
Line 37 - Please cite the proper research
Line 40 - Please replace “to” with “for”. Actually, the manuscript needs a careful proofreading
Line 47 - Narrow host range phages are not preferred over broader host range phages for use in phage therapy.
Line 55 - There is a direct connection between the content of phage’s genome and its therapeutic potential. But there is no such a connection between the classification scheme and its therapeutic potential. If you reclassify a phage, it will not change its biology or genomics.
Line 82 and Figure 1. These phage capsids does not look oblate but rather isometric.
Lines 91, 93. Do you mean “25 min” “20 min” and “5 min” or “0-25 min” “0-20 min” and “0-5 min”?
Section 2.2. Please provide the accession number or the supplementary file containing the phage genome in the GenBank format.
Line 152 - You could use HMM-based tools for a better annotation.
Line 157 Number of PG hydrolase families is much higher. Only E.coli contains about 35 such families.
Figure 6. Did you take into account the coverage? I would recommend using VIRIDIC instead of orthoANI
Section 2.3. Please classify your phage according to https://www.ncbi.nlm.nih.gov/pmc/articles/PMC8003253/. I also see that ICTV does not classify many temperate taxa at family level. But you could consider a new genus.
Section “Materials and Methods” - Please describe how you assemble the phage genome.
Reviewer 3 Report
Comments and Suggestions for Authors
The manuscript is well-researched and well-written; however, the authors could make several improvements:
- The family and genus names of phages should be italicized.
- In line 94, Vibrio should be italicized.
- For Figures B, C, and D, the data should be presented as bar graphs, as they represent different sets of phages.
- In Figure 3, the title should be revised to "Inhibition Curve," as a "Killing Curve" typically involves bacterial colony counting.
- The authors should include the complete genome annotation results in a supplementary table.
- How were the results in Figure 6A analyzed? The authors should clarify the methodology.
- How is a new viral family classified?
Comments on the Quality of English Language
- Typos can be detected.
Round 2
Reviewer 2 Report
Comments and Suggestions for Authors
Line 157 - "belongs to one of the four families of peptidoglycan hydrolases" - As I noted befor, here are much more than 4 PG gamilies. Please check and revise.
Line 63 and elsewhere - All viral taxa including Duplodnaviria, Heunggongvirae, Caudoviricetes should be written italics. Please revise.
Other comments have been addressed, thank you.
